# Use of Anti-Cytokine Therapy in Kidney Transplant Recipients with COVID-19

**DOI:** 10.3390/jcm10081551

**Published:** 2021-04-07

**Authors:** Marta Bodro, Frederic Cofan, Jose Ríos, Sabina Herrera, Laura Linares, María Angeles Marcos, Alex Soriano, Asunción Moreno, Fritz Diekmann

**Affiliations:** 1Infectious Diseased Department, Institut d’Investigacions Biomèdiques August Pi i Sunyer (IDIBAPS), University of Barcelona and Hospital Clinic, 08036 Barcelona, Spain; sherrera@clinic.cat (S.H.); lalinares@clinic.cat (L.L.); asoriano@clinic.cat (A.S.); amoreno@clinic.cat (A.M.); 2Department of Nephrology and Renal Transplantation, Institut d’Investigacions Biomèdiques August Pi i Sunyer (IDIBAPS), University of Barcelona and Hospital Clinic, 08036 Barcelona, Spain; fcofan@clinic.cat (F.C.); fdiekman@clinic.cat (F.D.); 3Medical Statistics Core Facility, Biostatistics Unit, Faculty of Medicine, Universitat Autònoma de Barcelona, 08193 Bellaterra, Spain; jose.rios@uab.cat; 4Microbiology Department, Institut d’Investigacions Biomèdiques August Pi i Sunyer (IDIBAPS), University of Barcelona and Hospital Clinic, 08036 Barcelona, Spain; mmarcos@clinic.cat

**Keywords:** COVID-19, kidney transplantation, anti-cytokine therapy, mortality, infection

## Abstract

In the context of the coronavirus disease 2019 (COVID-19) pandemic, we aimed to evaluate the impact of anti-cytokine therapies (AT) in kidney transplant recipients requiring hospitalization due to severe acute respiratory syndrome coronavirus 2 (SARS-CoV-2) infection. This is an observational retrospective study, which included patients from March to May 2020. An inverse probability of treatment weighting from a propensity score to receive AT was used in all statistical analyses, and we applied a bootstrap procedure in order to calculate an estimation of the 2.5th and 97.5th percentiles of odds ratio (OR). outcomes were measured using an ordinal scale determination (OSD). A total of 33 kidney recipients required hospitalization and 54% of them received at least one AT, mainly tocilizumab (42%), followed by anakinra (12%). There was no statistical effect in terms of intensive care unit (ICU) admission, respiratory secondary infections (35% vs. 7%) or mortality (16% vs. 13%) comparing patients that received AT with those who did not. Nevertheless, patients who received AT presented better outcomes during hospitalization in terms of OSD ≥5 ((OR 0.31; 2.5th, 97.5th percentiles (0.10; 0.72)). These analyses indicate, as a plausible hypothesis, that the use of AT in kidney transplant recipients presenting with COVID-19 could be beneficial, even though multicenter randomized control trials using these therapies in transplanted patients are needed.

## 1. Introduction

Severe acute respiratory syndrome coronavirus 2 (SARS-CoV-2) was first identified in December 2019 in Wuhan, China, and this novel coronavirus caused a national outbreak of severe pneumonia (coronavirus disease 2019 (COVID-19)) in China, rapidly spreading around the world thereafter, with more than 100,000,000 confirmed cases [1]. 

SARS-CoV-2 causes respiratory symptoms similar to those reported for SARS-CoV and Middle East Respiratory Syndrome Coronavirus (MERS-CoV). According to data from Wuhan local hospitals, and confirmed after in the rest of the world, the most common symptoms of COVID-19 were fever and dry cough at the onset of illness [2,3,4]. However, the most characteristic symptom of patients is respiratory distress, with many requiring intensive care management [2]. Moreover, accumulating evidence suggests that a subgroup of patients with severe COVID-19 have a cytokine storm syndrome, characterized by increased interleukin (IL)-1, IL-6, IL-7, interferon-γ inducible protein 10 among others [3,5]. Importantly, host inflammatory responses appear to constitute an important cause of associated organ injury [5] and anti-cytokine therapies have been postulated as potential therapeutic options [6].

Some data have recently been published among solid organ transplant (SOT) recipients presenting with COVID-19 [7,8,9,10,11,12]. Nevertheless, information regarding use of biological therapies in this population is scarce and what is more, potential side effects of biologics are secondary infections, including opportunistic infections, which would add a pre-existing risk due to immunosuppressive therapy are lacking. 

In this setting, we aimed to analyze kidney transplant recipients who required hospitalization due to COVID-19 and to evaluate the impact of anti-cytokine therapies on the outcomes.

## 2. Materials and Methods

From 6 March to 24 May, all kidney transplant recipients with respiratory symptoms and radiological evidence of pneumonia were admitted to the Hospital Clínic of Barcelona in the context of SARS-CoV-2 pandemic. Definitive diagnostic was established by a positive polymerase-chain reaction (PCR) from a nasopharyngeal swab. Clinical criteria for defining a case of SARS-CoV2 were the presence of respiratory symptoms with uni or bilateral interstitial infiltrates in the chest X ray. 

### 2.1. Management of Coronavirus Disease 2019 (COVID-19) 

In the first instance, our hospital protocol consisted of lopinavir/ritonavir 400/100 mg twice a day (BID) for 7–14 days plus hydroxychloroquine 400 mg/12 h on the first day, followed by 200 mg/12 h for the next 4 days. From 18 March, azithromycin 500 mg for 24 h and 250 mg/24 h for 4 additional days was added to the protocol. All patients with risk factors for thrombosis received prophylactic doses of low-weight heparin. The local indication for anti-cytokine therapy was restricted for patients with pneumonia, progressive respiratory failure (increasing fraction of inspired oxygen) and C-reactive protein (CRP) ≥8 mg/dL or ferritin ≥800 ng/mL or lymphocyte count <800 cells/mm^3^. The choice of anti-cytokine therapy was taken at the discretion of the attending physician. The dose of tocilizumab was 400 mg/24 h iv for patients with ≤75 kg and 600 mg/24 h intravenous (iv) for those with >75 kg, patients with no improvement could receive additional doses every 12 h up to a maximum of 3 doses. The dose of anakinra was 200 mg/12 h sc for 24 h and 200 mg/24 h with a maximum of 3 doses. The dose of baricitinib was 4 mg/24 h with a maximum of 4 doses. High doses of metilprednisolone (250 mg/24 h for 3 days followed by 30 mg/24 h for 3 days) could be prescribed in patients presenting with poor outcomes despite receiving biologic therapy. Hepatitis B serologies (hepatitis B surface antigen) and QuantiFERON-TB^®^ was performed prior to anti-cytokine prescription and prophylaxis with entecavir and isoniazid respectively were individually assessed. Secondary infections were defined as infections that occur during or after COVID-19. If they occurred 48 h after admission they were defined as hospital-acquired superinfections, whereas if they were diagnosed at the time of or within the first 24 h of hospital admission they were defined as community-acquired co-infections. We used the RIFLE criteria [13] (Risk, Injury, Failure, Loss, and End-stage kidney disease) to define impaired kidney graft function. Additionally, an increase in serum creatinine of 1.5–2 times, or a decrease in the glomerular filtration rate of more than 25% over baseline, was also considered in the definition of impaired kidney graft function. 

### 2.2. Management of Immunosuppressive Treatment 

According to our center’s policy, due to the potential severity of SARS-CoV-2 infection, mycophenolate and mTOR inhibitor (sirolimus or everolimus) were initially withdrawn in all admitted kidney transplant recipients with COVID-19. Furthermore, in patients starting treatment with lopinavir/ritonavir, the calcineurin inhibitor (CNI) (tacrolimus or cyclosporine) was also temporarily discontinued due to the strong interactions resulting in the increase of CNI levels. Maintenance immunosuppression consisted of prednisone monotherapy (20 mg/day) until COVID-19 resolution, at which time tacrolimus was reinitiated at reduced doses (through blood levels around 5 ng/mL).

### 2.3. Ordinary Scale Determination (OSD)

We used a clinical ordinary scale determination (OSD) to assess patient clinical status. This OSD was recorded at baseline and during hospitalization. The ordinal scale categories were: (1) Patients ready for discharge, (2) Patients requiring non-intensive care unit (ICU) hospital ward not requiring supplemental oxygen, (3) Patients requiring non-ICU hospital ward requiring supplemental oxygen, (4) Patients hospitalized in ICU or non-ICU hospital ward, requiring non-invasive ventilation or high-flow oxygen, (5) Patients hospitalized in ICU requiring intubation and mechanical ventilation, (6) Patients hospitalized in ICU, requiring extracorporeal membrane oxygenation (ECMO) or mechanical ventilation and additional organ support (e.g., vasopressors, renal replacement therapy) and (7) Death. 

### 2.4. Statistical Analysis

Results are shown as median and interquartile range (IQR: 25th and 75th percentiles) or absolute frequencies and percentages for quantitative and qualitative variable respectively. 

Probability of poor clinical outcome, defined as OSD >= 5, was estimated by means the odds ratio (OR) and their 95% confidence intervals (95%CI) from a weighted logistic regression models using the inverse of probability of treatment weighting (IPTW). This IPTW was used as a weight in order to create a synthetic sample with the distribution of covariates independent of biological prescription [14]. 

This IPTW was derived from a propensity score (PS) to receive biologic treatment from the following parameters: age, sex, number of comorbidities, basal creatinine, number of analytical values in last tercile, to establish high analytical alteration, prior transplantation, days from symptom onset to test, type of immunosuppressive regimen, baseline OSD and therapeutic effort limitation. Finally, this IPTW was stabilized by proportion of prescription to biological treatment. 

We calculated standardized differences, as differences between groups divided by pooled standard deviation, to assess homogeneity between patients with biologic prescription or not in their baseline characteristics. After IPTW use, some authors consider the cut-off point for standardized differences to be at ±0.20 [15], in this study all covariates were well balanced with the exception of a baseline result of OSD > 2, this particular misbalance is probably due to the fact that only one patient without biologic treatment had a baseline OSD > 2.

A bootstrap resampling procedure with replacement, with a rate of 80%, for 1000 samples, and a seed = 20200601 (date of closure of the database) was conducted as a measure of complementary results, in order to estimated 2.5th and 97.5th percentiles of OR. Missing data imputation for D-dimer, C-reactive protein, lactate dehydrogenase (LDH), ferritin and lymphocytes count was undertaken using the expectation-maximization (EM) algorithm [16] which relies on the flexible and reasonable missing at random (MAR) assumption, using age, sex, therapeutic effort limitation, secondary infection, hypertension, diabetes, cardiopathy and pneumopathy as additional factors for imputation. The amount of imputed missing data was 3% for D-dimer and lymphocytes, 9% for LDH and 12% for ferritin.

In all statistical analyses we applied a two-sided type I error of 5%. SPSS v.25 (IBM) and SAS v9.4 (Cary, NC, USA) were used for the analysis.

## 3. Results

During the study period, 1742 patients were hospitalized due to COVID-19 in our hospital and 33 of them were kidney transplant recipients. 

Baseline characteristics of kidney transplant recipients with SARS-CoV2 infection are described in Table 1. Sixty one percent of the kidney transplant recipients were male, and most of them had arterial hypertension (91%) and received non-mTOR-inhibitor based regimen (64%) (mainly tacrolimus-mycophenolic acid) and 36% an mTOR-inhibitor based regimen (sirolimus or everolimus). Twenty-one percent of them had received a previous transplant, and none of them had a history of acute rejection episode in the preceding 3 months. Median time from transplantation to diagnosis was 5.5 years (IQR 0.5; 21) and median time from symptom onset to positive test was 6 days (IQR 0; 20). Hydroxicloroquine was prescribed in 91% of patients, azithromycin in 85% and lopinavir/ritonavir in 82%. Forty-two percent of patients received tocilizumab (14/33), 24% high doses of steroids (8/33), 18% anakinra (6/33) and 3% baricitinib (1/33). Three patients (9%) received tocilizumab and anakinra. All patients were discharged at the end of the study period. 

Table 2 shows the description of baseline characteristics of patients stratified by biologic prescription and clinical presentation of kidney transplant recipients with COVID-19. Standardized differences showed baseline heterogeneity between both groups for number of analytical results in last tercile. Biologic treatment was prescribed more frequently in patients presenting with respiratory insufficiency compared with those with no respiratory insufficiency (74% vs. 26%). Six patients required ICU admission in the biologic group (33%). Two of these patients required ICU admission >24 h after biologic infusion, but none of them required invasive mechanical ventilation. No immediate drug-related side effects such as severe neutropenia, thrombocytopenia or hepatitis were reported. 

Seven patients presented a secondary respiratory infection and 3 of them had received both biologic and high dose steroids treatment, 3 only anti-cytokine therapy and the remaining patient received high dose steroids. All infections were of bacterial etiology except one, which was fungal (invasive aspergillosis). Seven patients in our cohort presented a secondary respiratory infection; 3 of them had received both biologic and high dose steroid treatment, 3 had received only anti-cytokine therapy and the remaining patient had received high-dose steroids only. All infections were bacterial with one exception that was fungal (invasive aspergillosis). Four of them required ICU admission, one required extracorporeal membrane oxygenation and 3 of them required non-invasive ventilation. Those admitted to the ICU had additional secondary respiratory infections during their ICU stay. The median number of days from biologic therapy to secondary infection was 10 days (IQR 5–14). All patients survived except one (who needed extracorporeal membrane oxygenation therapy). 

Weighted logistic regression analysis by IPTW for outcomes in terms of OSD by biologic prescription is described in Table 3. Use of biologic therapy was associated with a minor probability of a maximum OSD index during hospitalization ≥5 (OR 0.17 (95%CI: 0.01; 3.83)) without initial statistical significance, but after performing simulation by bootstrap, 2.5th and 97.5th percentiles of calculated OR were between 0.10 and 0.72 suggesting a possible statistical association in further studies. 

The weighted analysis was made with an ITPW from a propensity score to (PS) to received biologic from the following parameters: age, sex, number of comorbidities, basal creatinine, number of analytical values in last tercile, prior transplantation, days from symptom onset to test, type of immunosuppressive regimen, baseline OSD and therapeutic effort limitation. Due to the limitation of sample size, a bootstrap resampling (*n* = 1000) with replacement was performed and we presented 2.5th and 97.5th of estimations of OR for these 1000 analyses.

## 4. Discussion

In this study we analyzed a cohort of kidney transplant recipients and evaluated the impact of anti-cytokine therapies use. Despite being a small sized-cohort; the simulation analysis with a resampling of 1000 samples suggests that it would be possible to conclude, as a hypothesis, a potential beneficial effect of the use of AT in kidney transplant recipients. 

Accumulating evidence suggests that the host’s immune response and development of tissue-focused inflammation in the lung likely play an important role in COVID-19 pathogenesis [17]. Patients with severe COVID-19 can have a cytokine storm syndrome characterised by increased interleukin (IL)-1, IL-2, IL-6, granulocyte-colony stimulating factor, interferon-Ɣ inducible protein 10, macrophage inflammatory protein 1α and tumor necrosis factor (TNF)-α [3] and elevated serum concentrations of Il-1β, IL-6 and other inflammatory cytokines; they are are hallmarks of severe forms [18,19,20]. Elevated serum C-reactive protein, a protein whose expression is driven by IL-6, is also a biomarker of severe infection. In line with this, patients from our study showed high levels of C-reactive protein and ferritin, other parameters that have been associated with the inflammatory cytokine storm related to COVID-19 [19]. We speculated that patients with severe clinical manifestations and elevated laboratory parameters of inflammation could benefit from anti-cytokine therapies, and thus these therapies were introduced in our centre protocol for selected patients. 

Tocilizumab was the most frequently prescribed biologic drug in this group of patients. It is a recombinant humanized anti-IL-6 receptor monoclonal antibody which is approved for the treatment in rheumatologic diseases [21]. Additionally, it has been used in some cases of chimeric antigen receptor (CAR) T-cell therapy induced cytokine storm and secondary encephalopathy with favourable outcomes [22]. In kidney transplantation, tocilizumab has been used as rescue therapy in patients with chronic antibody-mediated rejection (ABMR) who failed standard-of-care treatment with no significant adverse events [23]. Recent studies have found a beneficial effect of tocilizumab in reducing mortality, need for mechanical ventilation and shortening hospitalization [24,25], however, other studies found no benefit in the use of tocilizumab [26]. Anakinra, the second most frequently prescribed biologic treatment in this group of patients, is an IL-1 receptor antagonist with a very safe profile [27]. It is the cornerstone treatment for hyperinflammatory conditions such as Still’s disease, and also has been shown to be highly effective in the treatment of cytokine storm syndromes, including macrophage activation syndrome and cytokine release syndrome [28]. It has been used as a safe and effective therapeutic option for gout in patients with chronic kidney disease [29]. In a non-SOT population presenting with COVID-19, anakinra had shown promising results [30,31,32]. Finally, baricitinib is a small molecule, orally administered, JAK-1 and -2 selective inhibitor used in patients with moderate or severe rheumatoid arthritis or patients with other active disease-modifying antirheumatic drugs with inadequate responses to prior therapies [33]. Studies evaluating the use of baricitinib as COVID-19 therapy are still ongoing and preliminary results showed potential benefits [34]. 

We found that the mortality rate of kidney transplant-recipients was similar to that of non-SOT patients [35] and similar to other cohorts of COVID-19 SOT patients [8,9]. One could expect some protective effect of immunosuppressive therapies from the inflammatory storm or even due to the in vitro activity shown by some drugs such as cyclosporine, tacrolimus and mTOR inhibitors [18]. However, median years from transplantation to COVID-19 in our study were 5.5, which was not in the peak period of immunosupression and moreover, calcineurin inhibitors and mTOR inhibitors were mostly suspended during hospitalization due to drug interactions with lopinavir/ritonavir. 

The strengths of this clinical study are that the cohort included all cases of COVID-19 kidney transplant recipients during the study period that received AT with promising results. However, it has some limitations. First of all, as it is a single-centre study, our findings may be attributable to institution-specific variables and may not reflect the epidemiology of different centers and/or geographical areas. Secondly, the size of this cohort is too small to draw strong conclusions and, therefore, randomized controlled trials evaluating the impact of the use biologic treatments in immunosuppressed patients are imperative. Furthermore, the real incidence of SARS-CoV-2 infection in the SOT population is unknown and we might be underestimating the real impact. Nonetheless, all kidney transplant recipients are periodically and closely followed-up, even during the pandemic period. We are planning to perform serologic tests in all recipients to make a better estimate of the real incidence of SARS-CoV-2 infection in this population. Finally and importantly, the study was performed during the first wave of the COVID-19 pandemic, and the standard of care treatment has changed since then as several randomized trials have shown the benefit of some drugs such as remdesivir and dexamethasone [36,37,38], which are currently used.

## 5. Conclusions

To conclude, we found that the use of AT in our cohort of kidney transplant recipients was safe and responses, in terms of clinical efficacy using the OSD score, were favorable. However, independent studies in order to confirm these findings and randomized clinical trials evaluating the impact of using AT in SOT recipients over the long term are needed. 

## Figures and Tables

**Table 1 jcm-10-01551-t001:** Baseline characteristics of kidney transplant recipients hospitalized due to coronavirus disease 2019 (COVID).

Variable	Total *n* (%)
Age, median, IQR	55 (33–86)
Male sex	20 (61)
Hypertension	30 (91)
Diabetes mellitus	9 (27)
Cardiopathy	11 (33)
Chronic obstructive pulmonary disease	4 (12)
Cause of end- stage kidney disease ▪ Nephroangiosclerosis ▪ Glomerulonephritis ▪ Diabetes mellitus ▪ Polycystic kidney disease ▪ Ureteral reflux ▪ Not diagnosed	
	5 (15)
	10 (30)
	1 (3)
	6 (18)
	2 (6)
	9 (27)
Prior transplantation	7 (21)
Multivisceral transplantation	2 (6)
Immunosupressive regimen ▪ Non mTOR based regimen ▪ mTOR based regimen	
	21 (64)
	12 (36)
Acute allograft rejection (3 months prior)	0
Years from transplant to diagnosis, median, IQR	5.5 (0.5–21)
Presenting symptoms ▪ Fever ▪ Fatigue ▪ Hyposmia/ageusia ▪ Cough ▪ Diarrhea	
	29 (91)
	3 (9)
	3 (9)
	18 (56)
	6 (18)
Days from symptom onset to test, median, IQR	6 (0–20)
Serum basal creatinine (mg/dL), IQR	1.4 (0.8–3.7)
Impaired kidney graft function	17 (52)
Respiratory insufficiency	21 (64)
ICU admission	11 (33)
Invasive mechanical ventilation	4 (12)
Maximum ferritin levels (ng/mL) (NR) 20–400	888 (281–4372)
Maximum C-reactive protein (mg/dL) NR < 0.4	14 (0–25)
Maximum Lactate dehydrogenase (U/L) NR < 234	254 (198–687)
Minimum lymphocytes count (1000/mm^3^)	600 (100–1000)
D-dimer (ng/mL) NR < 500	1300 (500–12400)
Secondary infection	7 (23)
Median days of hospitalization (IQR)	12 (4–59)
Overall mortality	4 (12)

Interquartile range (IQR); normal range (NR).

**Table 2 jcm-10-01551-t002:** Baseline characteristics and clinical presentation of kidney recipients with COVID-19 by biologic therapy prescription and standardized difference depending on before and after application of inverse of probability of treatment weighting (IPTW) in the comparison.

Variable	Biologic (*n* = 19)	Non Biologic (*n* = 14)	*p* Value	Standarized Difference Raw IPTW Adjusted	
Age, median, IQR	52 (37; 83)	61 (33; 87)	0.4	−0.070	−0.141
Male sex	11 (61)	9 (60)	1	0.131	−0.080
Hypertension	15 (83)	15 (100)	0.1		
Diabetes mellitus	5 (28)	4 (27)	1		
Cardiopathy	6 (33)	5 (33)	1		
Chronic obstructive pulmonary disease	4 (22)	0	0.1		
Number of comorbidities, median(IQR)	1 (0; 4)	1.5 (1; 3)	0.6	0.197	0.158
Prior transplantation	4 (22)	3 (20)	1	−0.009	0.033
Multivisceral transplantation	2 (11)	0	0.5		
Immunosupressive regimen ▪ Non-mTOR-inhibitor based ▪ mTOR-inhibitor based			1	0.0235	−0.034
	11	10			
	7	5			
Years from transplant to diagnosis, median (IQR)	4.8 (0.5; 15.5)	6.2 (0.5; 21.6)	0.4		
Days from symptom onset to test, median (IQR)	6 (1; 20)	6 (2; 15)	0.7		
Days from symptom onset to test <7	6 (31.6)	4 (28.6)		−0.009	0.0326
Baseline OSD, median, IQR	2 (2; 4)	2 (2; 5)	0.09		
Baseline OSD >2 (needed O_2_)	6 (31.6)	1 (7.1)		0.6503	0.3035
OSD during hospitalization, median, IQR	3 (2; 6)	2 (2; 5)	0.004		
Serum basal creatinine	1.5 (0.8; 3.4)	1.3 (0.8; 3.7)	0.3	0.2311	0.0457
Impaired kidney graft function	11 (58)	6 (46)	1		
Maximum ferritin levels (ng/mL) NR 20–400	1056 (300; 4372)	361 (281; 3281)	0.5		
Maximum C-reactive protein (mg/dL) NR < 0.4	15 (4; 26)	11 (0; 20)	0.09		
Maximum lactate dehydrogenase (U/L) NR < 234	367 (267; 687)	276 (198; 562)	0.1		
Minimum lymphocytes count (1000/mm^3^)	600 (100; 1000)	500 (200; 800)	0.8		
D-dimer (ng/mL) NR < 500	1300 (1000; 12400)	1300 (500; 10000)	1		
Number of analytic results in last tercile	1 (1; 3)	2 (1; 3)		0.6692	0.1008
Respiratory insufficiency	14 (74)	5 (26)	0.03		
ICU admission	6 (33)	4 (27)	0.7		
ICU admission post biologic infusion (>24 h)	2 (11)	-	-		
Invasive mechanical ventilation	2 (11)	1 (7)	1		
Invasive mechanical ventilation post biologic infusion (>24 h)	0	-			
Treatment received ▪ lovinavir/ritonavir ▪ azithromycin ▪ hydroxychloroquine ▪ high-dose steroids					
	18 (100)	9 (69)	0.02		
	17 (94)	11 (85)	0.6		
	18 (100)	12 (92)	0.4		
	3 (17)	0 (7)	0.2		
Secondary infection	6 (35)	1 (7)	0.1		
Overall mortality	3 (16)	2 (13)	1		
Therapeutic effort limitation	2 (10.5)	1 (7.1)	1	0.1194	0.0414

Interquartile range (IQR); ordinary scale determination (OSD); normal range (NR); intensive care unit (ICU).

**Table 3 jcm-10-01551-t003:** Logistic regression analysis with IPTW for outcomes in terms of OSD by biologic prescription.

	Logistic Regression w/IPTW		Logistic Regression w/IPTW and Bootstrap (*n* = 1000) w/Replacement
	OR (95%CI)	*p*-value	OR (2.5; 97.5 percentiles)
OSD = 7	0.66 (0.07; 6.60)	0.7254	0.75 (0.15; 2.16)
OSD during hospitalization ≥ 5	0.17 (0.01; 3.83)	0.2669	0.31 (0.10; 0.72)
Maximum OSD ≥ 5	0.41 (0.05; 3.38)	0.4089	0.54 (0.11; 1.72)

Ordinary scale determination (OSD).

## Data Availability

Data supporting these results is available under request.

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
