# Peer review of "Use of Anti-Cytokine Therapy in Kidney Transplant Recipients with COVID-19"

_jcm, 2021, doi:10.3390/jcm10081551_

Round 1
Reviewer 1 Report
This is an interesting study investigating the use of biological agents in kidney transplant recipients.
1) Line 22 of the abstract: "Nevertheless, received a AT drug..." This phrase should be rewritten in a more correct and comprehensive manner
2) Line 44 of the introduction: Several more observational studies from large registries regarding COVID-19 in kidney transplant recipients have been published since June 2020.
3) Perhaps you should address in the discussion section results from previous studies regarding the use of biological agents in COVID-19 in SOT recipients and the general population (randomized trials).
4) The study includes kidney transplant recipients during the 1st wave of the COVID-19 pandemic. The standard of care treatment has changed since then (eg dexamethasone). Perhaps, this could be included in the limitations of the study.
Reviewer 2 Report
The article „Use of anti-cytokine therapy in kidney transplant recipients with COVID-19“ written by Marta Bodro and colleagues focusses on a an urging problem in transplantation medicine: how to treat SOT patients suffering from COVID-19.
However some relevant aspects need/should be addressed:
1) Regarding subitem “Management of COVID-19”: the authors explain their local protocol and local indication for anti-cytokine therapy: which rationale caused the decision for the specific anti-cytokine therapy, e.g. either tocilizumab, or kineret,.. or was this done by chance? Did the authors measure IL-6 and did it differ in treated and non-treated pts?; did the authors adapt anti-cytokine therapy eg. anakinra to existing kidney function? Or did they used the given drugs in a fixed manner?
2)what was the BMI of the kidney transplant recipients and did it differ between both groups? How was impaired kidney function defined?
3) had the usage of anti-cytokine therapy any impact regarding length of invasive mechanical ventilation, ICU or in-hospital stay;or time till death?
4) did treated and non-treated pts differ regarding time till COVID-19 virus clearance? and had used anti-cytokine therapy any impact on SARS-CoV-2 specific antibody formation?
5)Besides postulated beneficial effects the detected side effects - mainly infections - in the pts, treated with anti-cytokine therapy (n = 6; 35%) had to keep in mind; can the authors state on the severity of these secondary infections as well as on the duration resp. timely relationship to ant-cytokine protocols.
Overall - besides given logistic regression analysis - one has to be carefull, what drawn conclusions are not over overrated, because for clear evidence of beneficial effects of anti-cytokine therapy randomized controlled trialy are mandatory
Round 2
Reviewer 2 Report
see below